# Drug Resistance in Colorectal Cancer: From Mechanism to Clinic

**DOI:** 10.3390/cancers14122928

**Published:** 2022-06-14

**Authors:** Qianyu Wang, Xiaofei Shen, Gang Chen, Junfeng Du

**Affiliations:** 1The Second School of Clinical Medical, Shanxi Medical University, Taiyuan 030001, China; wangqy9264@163.com; 2Department of General Surgery, Affiliated Drum Tower Hospital of Nanjing University Medical School, Nanjing 210008, China; dg1535058@smail.nju.edu.cn; 3Medical Department of General Surgery, The 1st Medical Center, Chinese PLA General Hospital, Beijing 100853, China; 4Department of General Surgery, The 7th Medical Center, Chinese PLA General Hospital, Beijing 100700, China; 5The Second School of Clinical Medicine, Southern Medical University, Guangzhou 510515, China

**Keywords:** colorectal cancer, drug resistance, chemotherapeutic agents, molecularly targeted therapy, immunotherapy, reversal strategies

## Abstract

**Simple Summary:**

Chemotherapy, radiotherapy and molecularly targeted therapy could improve the prognosis of colorectal cancer (CRC) patients. Recently, immunotherapy, especially immune checkpoint inhibitors, has significantly improved the prognosis of some patients. However, drug resistance significantly reduces the usefulness of these drugs. Although research on the molecular mechanisms underlying the emergence of drug resistance is ongoing, the specific molecular mechanisms remain unclear. This article reviews the findings on the mechanisms of drug resistance in CRC patients in preclinical and clinical studies, which may provide valuable directions for future in-depth study of drug resistance.

**Abstract:**

Colorectal cancer (CRC) is one of the leading causes of death worldwide. The 5-year survival rate is 90% for patients with early CRC, 70% for patients with locally advanced CRC, and 15% for patients with metastatic CRC (mCRC). In fact, most CRC patients are at an advanced stage at the time of diagnosis. Although chemotherapy, molecularly targeted therapy and immunotherapy have significantly improved patient survival, some patients are initially insensitive to these drugs or initially sensitive but quickly become insensitive, and the emergence of such primary and secondary drug resistance is a significant clinical challenge. The most direct cause of resistance is the aberrant anti-tumor drug metabolism, transportation or target. With more in-depth research, it is found that cell death pathways, carcinogenic signals, compensation feedback loop signal pathways and tumor immune microenvironment also play essential roles in the drug resistance mechanism. Here, we assess the current major mechanisms of CRC resistance and describe potential therapeutic interventions.

## 1. Introduction

Colorectal cancer (CRC) is the third most common and second deadliest malignancy in the world [1]. Over the past decade, the development and use of chemotherapy, molecularly targeted therapy and immunotherapy have significantly improved patient outcomes (Figure 1). These anti-cancer drugs are first metabolized into an active form, then transported to the appropriate location and used on the appropriate target, ultimately leading to tumor cell death. The aberration of these processes could lead to the emergence of drug resistance in cancer treatment and significantly reduce its utility. Here, we not only explain the role of the pathway mentioned above that is aberrant in drug resistance, but also highlight the influence of compensatory feedback loop signal path and tumor microenvironment (TME) on CRC resistance, thus providing possible strategies for reversing drug resistance in CRC (Table 1 and Table 2).

## 2. Drug Metabolism

Only when cancer drugs are transformed into active forms can they maximize their anti-tumor effects. Thus, reduced activity or increased degradation of anti-cancer drugs would lead to drug resistance (Figure 2).

### 2.1. Reduced Drug Activation

Capecitabine (CAP) needs to be transformed into 5-fluorouracil (5-FU) for direct cytotoxicity, which is mediated by thymine phosphorylase (TP) [2]. Preclinical studies show that methylation of extracellular growth factor-1 (ECGF-1 gene, encoding TP protein) inhibits TP gene transcription and leads to drug resistance to CAP, while DNA methyltransferase (DNMT) inhibitors can reverse drug resistance to CAP [3].

5-FU exerts its anti-cancer effects by inhibiting thymidylate synthase (TS) and incorporating its metabolites into RNA and DNA [4]. Orotate phosphoribosyl-transferase (OPRT), uridine monophosphate (UMP) synthetase (UMPS) and UMP kinase (UMPK) are responsible for the conversion of 5-FU into active anti-cancer metabolites in tumor cells [5]. It has been proven that the lower expression of these metabolic enzymes is associated with 5-FU resistance in CRC patients [6,7,8]. In addition, TP can convert 5-FU into 5-fluoro-2′-deoxyuridine (FdU), which is a form that converts 5-FU into an active form. Studies have shown that the higher expression of TP predicts a good response to 5-FU [9,10]. Although another study has shown that there is no correlation between the level of TP expression and tumor response rate, the progression-free survival of patients with higher expression of TP is significantly prolonged [11]. These data suggest that patients with high expression of these enzymes that convert 5-FU to the active form may be more likely to benefit from 5-FU.

**Table 1 cancers-14-02928-t001:** Promising strategies for reversing drug resistance in preclinical studies.

Therapy	Treatment Regimen	Target	Subpopulation	Model System	Reference
ABC inhibitors	Cryptotanshinone/dihydrotanshinone/cinobufagin/8-oxocoptisine	P-gp	/	Cell	[12,13,14]
	Liposomal antisense oligonucleotides	P-gp, MRP1, MRP2, Bcl-2, Bcl-XL	/	Cell	[15,16]
	Y-box binding protein 1 (YB-1)	TOP-1	/	Cell	[17].
New anti-EGFR mAbs	Sym004	EGFR ECD	EGFR G465E mutant	Cell/mice	[18]
	MM-151	EGFR ECD	EGFR G465E mutant	PDX	[19]
	Necitumumab	EGFR ECD	EGFR S468R mutant	Cell	[20]
MEK inhibitors	BAY86-9766 + cetuximab	MEK	KRAS mutant	PDX	[21]
	Pimasertib + cetuximab	MEK	KRAS, NRAS or BRAF mutant	Cell/mice/PDX	[22]
	Trametinib + Olaparib + αPD-L1	MEK	KRAS mutant	Mice	[23]
	G-38963 + αPD-1	MEK	KRAS mutant	Mice	[24]
New anti-MEK mAbs	SCH772984	MEK	KRAS or BRAF mutant	PDX	[25]
IGF-IR inhibitors	NVP-AEW541(IGF-IR i) + aEGFR	IGF-IR	/	CRC primary cell line	[26]
HER2 mABs	Pertuzumab + cetuximab/lapatinib	HER2	HER2 amplification	PDX	[27]
	Trastuzumab + cetuximab	HER2	/	Cell	[28]
	4D5 + cetuximab	HER2	/	PDX	[29]
HER3 mABs	U3-1402	HER3	High HER3 expression;	CDX	[30]
β-catenin/CBP inhibitors	TrpC5	β-catenin/CBP	/	Cell	[31]
	AQP5	β-catenin/CBP	/	Cell	[32]
	ICG-001	β-catenin/CBP	/	Cell/mice	[32]
TLR9 agonist	SD-101 + PD-1	TLR-9	/	Mice	[33]
	IMO + cetuximab/bevacizumab	TLR-9	/	Mice	[34]
	IMO + cetuximab + irinotecan	TLR-9	/	Cell and mice	[35]
	IMO + cetuximab	TLR-9	/	Cell and mice	[36]
TGF-β inhibitors	LY2157299 + Gefitinib	TGF-βIR	/	Cell	[37]
	PF-03446962 + Bevacizumab	ALK-1	/	Mice and PDX	[38]
	Galunisertib +αPD-L1	TGFBR1	low mutational burden, T-cell exclusion and TGFβ-activated stroma	Mice	[39]
Apoptotic inhibitors	ABT-263 + selumetinib	Bax, Bak	BRAF(V600E) or RAS mutant	Cell	[40]
	AEG35156 + taxanes	XIAP	/	Mice	[41]
	Birinapant + oxaliplatin/5-FU	XIAP	/	Cell	[42]
Autophagy inhibitors	Chloroquine/3-MA + 5-FU	Autophagy	/	Cell	[43,44]
	Chloroquine/3-MA + cetuximab	Autophagy	/	Cell	[45]
	3-MA + αEGFR	Autophagy	/	Cell	[46]
	SB02024/SAR405 + αPD-1/PD-L1	Vps34	/	Mice	[47]
Ferroptosis agonist	β-elemene + cetuximab	/	KRAS mutant	Cell and mice	[48]
	RSL3 + cisplatin	/	/	Cell and mice	[49]
CAF	BGJ398 + 5-FU + oxaliplatin	FGFR4	/	Cell	[50]
	BLU9931 + cetuximab	FGFR4	/	Cell and mice	[51]
	Regorafenib + cetuximab	FGFR, VEGF, PDGFR-β	/	Cell and mice	[52]
	GKT137831	NOX4	/	Mice	[53]
MDSC	KTN0158 + αPD-1 + αCTLA-4	KIT	/	Mice	[54]
TAM	DFMO + 5-FU	ODC	/	Mice	[55]
	RG7155	CSF1R	/	Mice	[56]
MDSC	R848 + oxaliplatin	TLR 7/8 agonist	/	Mice	[57]
MDSC and TAM	TP-16 + αPD-1	EP4	/	Mice	[58]
VEGF inhibitors	αVEGF(A) + αPD-1	VEGF-A	/	Mice	[59,60,61]
	Vargatef + afatinib	VEGF, EGFR	/	Mice	[62]
	Regorafenib + cetuximab	FGFR, VEGF, PDGFR-β	KRAS or BRAF mutant	Cell and mice	[52]

**Table 2 cancers-14-02928-t002:** Promising strategies for reversing drug resistance in clinical trials.

Therapy	Treatment Regimen	Target	Subpopulation	Species	Setting	Efficiency	Reference
New anti-EGFR mAbs	Sym004	EGFR ECD	/	anti-EGFR-refractory/unselected mCRC	Phase I trial studies	ORCD: (67%)	NCT01117428 [18]
	Sym004	EGFR ECD	KRAS WT	anti-EGFR-refractory mCRC	Phase II trial studies	mOS: (10.3 m:9.6 m)	2013-003829-29 [63]
	MM-151 + irinotecan vs. MM-151	EGFR ECD	KRAS WT	Refractory advanced CRC	Phase I trial studies	MM-151-SD: 31% MM-151+ irinotecan- PR: 1/3	NCT01520389 [64]
	Necitumumab	EGFR ECD	/	first-line treatment for locally advanced or metastatic CRC	Phase II trial studies	ORR: (63.6%) OS: (22.5 m)	[65]
BRAF inhibitor	Vemurafenib + panitumumab	BRAF V600E	BRAF-mutant	Refractory mCRC	Pilot trial	ORR: (16.7%)	[66]
	Vemurafenib + cetuximab vs. vemurafenib	BRAF V600E	BRAF V600E mutant	CRC	Phase II trial studies	ORR: (0%:4%)	NCT01524978 [66]
	Vemurafenib + irinotecan + cetuximab	BRAF V600E	BRAF V600E mutant	mCRC	Phase Ib trial studies	ORR: (35%)	NCT01787500 [67].
PI3K inhibitor	cetuximab+PX-866 vs. cetuximab	PI3K	KRAS WT	irinotecan- and oxaliplatin-refractory mCRC	Phase II trial studies	mOS: (266d:333d)	[68]
	Alpelisib + cetuximab + encorafenib + vs. cetuximab + encorafenib	PI3K	BRAF-Mutant	mCRC	Phase Ib trial studies	mOS: (4.2 m:3.7 m)	NCT0171938 [69]
IGF-IR inhibitor	IMC-A12 + cetuximab vs. IMC-A12	IGF-IR	/	anti-EGFR-refractory mCRC	Phase II trial studies	ORR: (0%:5%)	NCT00503685 [70]
	Dalotuzumab + Cetuximab + Irinotecan vs. Cetuximab + Irinotecan	IGF-IR	KRAS WT	chemo-refractory mCRC	Phase II/III trial studies	mOS: (10.8 m:14 m)	NCT00614393 [71]
HGF/IGF-IR inhibitor	Rilotumumab + panitumumab vs. ganitumab + panitumumab vs. panitumumab	HGF/IGF-IR	KRAS WT	previously treated mCRC	Phase Ib/II trial studies	mOS: (13.8 m:10.6 m:11.6 m)	NCT00788957 [72]
HER2 inhibitor	Neratinib + Cetuximab	HER2	Quadruple-WT(KRAS, NRAS, BRAF, PIK3CA)	anti-EGFR-refractory mCRC	Phase Ib trial studies	There were no objective responses	NCT01960023 [73]
	Trastuzumab + lapatinib	HER2	KRAS WT, HER2-positive	anti-EGFR-refractory mCRC	Phase II trial studies	ORR: (30%)	[74]
	Trastuzumab + Pertuzumab	HER2	HER2 amplification	treatment-refractory mCRC	Phase IIa trial studies	ORR: (32%)	NCT02091141 [75]
HER3 inhibitor	Lumretuzumab	HER3	HER3-positive	treatment-refractory mCRC	Phase I trial studies	SD: (21.3%)	NCT01482377 [76]
	MEHD7945A	HER3/EGFR	/	treatment-refractory mCRC	Phase I trial studies	SD: (28.6%)	NCT01207323 [77]
MET inhibitor	Tivantinib + Cetuximab	MET	KRAS WT	anti-EGFR-refractory mCRC	Phase II trial studies	ORR: (9.8%)	NCT01892527 [78]
	Tivantinib + irinotecan + cetuximab vs. irinotecan + cetuximab	MET	KRAS WT	locally advanced or metastatic CRC	Phase I/II trial studies	mPFS: (8.3 m:7.3 m) mOS: (19.8 m:16.9 m) ORR: 45%:33%	NCT01075048 [79]
	Capmatinib + cetuximab	MET	K/NRAS WT, MET-positive	anti-EGFR-refractory mCRC	Phase Ib trial studies	SD: (46.2%)	NCT02205398 [80]
VEGF/ALK-1 inhibitor	Regorafenib + PF-03446962	VEGF/ALK-1	/	treatment-refractory mCRC	Phase Ib trial studies	SD: (18.2%)	[81]
MEK inhibitor	Cobimetinib + atezolizumab	MEK	/	mCRC	Phase I/Ib trial studies	ORR: (8%)	NCT01988896 [82]
	Atezolizumab + cobimetinib vs. Atezolizumab vs. regorafenib	MEK	/	previously treated mCRC	Phase III trial studies	mPFS: (1.91 m:1.94 m:2.0 m) mOS: (8.87 m:7.10 m:8.51 m)	NCT02788279 [83]
	Cobimetinib + Atezolizumab + Hydroxychloroquine	MEK	KRAS-Mutant	Advanced Malignancies	Phase I/II trial studies	None	NCT04214418
CSF1R inhibitor	Pexidartinib + Durvalumab	CSF1R	/	Metastatic/Advanced CRC	Phase I trial studies	None	NCT02777710
	ARRY-382 + Pembrolizumab	CSF1R	/	Advanced Solid Tumors	Phase I/II trial studies	None	NCT02880371
anti-VEGF mAbs	Bevacizumab + Atezolizumab	VEGF-A	MSI-H	mCRC	Phase Ib trial studies	ORR: (30%) SD: (90%)	NCT01633970 [84]
	bevacizumab + capecitabine + atezolizumab vs. bevacizumab + capecitabine	VEGF-A	/	refractory mCRC	Phase II trial studies	mPFS: (3.3 m:4.4 m) ORR: (4.35%:8.45%) 12 m OS: (43%:52%)	NCT01633970 [85]
	nivolumab + standard of care chemotherapy + bevacizumab	VEGF-A	/	mCRC	Phase II/III trial studies	None	NCT03414983
anti-VEGFR mAbs	Regorafenib + cetuximab/Panitumumab	VEGFR	/	Unresectable, Locally Advanced, or Metastatic Colorectal Cancer	Phase I trial studies	Clinical Benefit: (53%)	NCT02095054 [86]
					Phase II trial studies	None	NCT04117945

Irinotecan exerts a cytotoxic effect by inhibiting DNA replication and transcription by targeting topoisomerase 1 (TOP-1) [87]. Carboxylesterase 2 (CES2) is the critical enzyme for the activation of irinotecan, which can catalyze the conversion of irinotecan to the more active form of SN-38 (a topoisomerase inhibitor) [88]. Multiple studies have shown that the expression of CES2 in CRC is positively correlated with irinotecan sensitivity [89,90]. Multiple preclinical studies have shown that selectively increasing the activity of CES in CRC by viruses can improve the efficacy of irinotecan [91,92,93]. Although some studies can indeed enhance the sensitivity of irinotecan in vitro and in vivo, this mechanism has not been studied in a clinical setting. In addition, it has been reported that genetically engineered human neural stem cells can increase CES activity in other tumors [94,95,96]. Gene therapy may be a future approach to increase the sensitivity of CRC to irinotecan chemotherapy.

### 2.2. Increased Drug Inactivation

The vast majority of 5-FU is catabolized into inactive metabolites by dihydropyrimidine dehydrogenase (DPD) in the liver [97]. As the name implies, high expression of DPD mRNA can confer resistance to 5-FU in CRC [9,10]. Therefore, interference with DPD is a way to overcome drug resistance in these patients. Preclinical studies have shown that methylation of the DPYD gene promoter, which encodes DPD protein, can inhibit DPYD transcription [98]. In addition, methylation of the DPYD promoter region can reduce the activity of DPD in clinical samples [99]. In addition, studies have reported that DPD inhibitors such as 5-ethynyluracil and 5-chloro-2,4-dihydroxypyridine (CDHP) can improve the anti-tumor activity of 5-FU to a greater extent [100,101,102]. However, patients with DPD deficiency will experience severe systemic poisoning symptoms when using 5-FU [103]. Therefore, the expression level of DPD should be further standardized to maximize the benefit of patients.

Irinotecan or its activated form SN38 can be metabolically inactivated by uridine diphosphate glucuronosyltransferase 1A1 (UGT1A1), cytochrome P450-3A4 (CYP3A4) and β-glucuronidase [104]. These enzymes involved in irinotecan metabolism are susceptible to environmental factors such as smoking, herbal supplements and drug therapy that alter the irinotecan metabolic pathway, resulting in altered irinotecan clearance [105,106,107]. Therefore, targeting these enzymes may be an effective way to reverse irinotecan resistance. Methimazole (CYP3A inhibitor) [108], neomycin (β-glucuronidase inhibitor) [109] and amoxapine (β-glucuronidase inhibitor) [110] have been shown to reduce SN-38 degradation effectively.

In addition, the transcription level of UGT1A1 is negatively regulated by DNA methylation [111]. Therefore, UGT1A1 promoter methylation and subsequent inhibition of UGT1A1-related metabolic pathways participate in the retention of active SN-38 in CRC, enhancing the therapeutic effect of irinotecan [112].

## 3. Drug Transport

When the membrane transporter pumps chemotherapeutic drugs or molecularly targeted drugs from the inside of the cell out of the cell, it would decrease the concentration of anti-tumor drugs in the tumor cell and the development of multi-drug resistance (MDR) [113]. To date, the human genome has been proven to have more than 400 membrane transporters, which are divided into two families: ATP binding cassette (ABC) and solute carrier (SLC) transporters [114]. Representative ABC transporters are multi-drug resistance protein 1 (MDR1; P-gp; ABCB1), breast cancer resistance protein (BCRP; ABCG2) and multi-drug-resistance-associated protein 1 (MRP1, ABCC1), while solute carrier (SLC) transporters include organic anion transporters, organic cation transporters and organic anion transport polypeptides [115]. In fact, ABC transporter is the main membrane transporter for MDR and plays an important role in CRC resistance to chemotherapy (Figure 2).

P-glycoprotein (P-gp) is the first identified ABC superfamily member and exists in various normal cells of the intestine [116]. P-gp is the most classical efflux pump mediating multi-drug resistance in tumor therapy; the ATP-driven hydrolysis of P-gp may pump chemotherapeutic drugs out of the cell. Preclinical and clinical studies have demonstrated that overexpression of P-gp in a variety of solid tumors, including CRC, leads to multi-drug resistance and chemotherapy failure. For example, P-gp is overexpressed in CRC cell lines and specimens resistant to 5-FU/Dox/oxaliplatin [31,117,118]. The overexpression of BCRP and MRP1 has subsequently been identified to be associated with drug resistance in various tumors, including CRC [119,120,121].

Given the vital position of ABC transporters in CRC resistance, inhibiting these transporters is an effective way to reverse resistance. Therefore, ABC transporter inhibitors have emerged. Since the emergence of the first-generation P-gp inhibitor verapamil in 1981, P-gp inhibitors have been studied in depth, and the third-generation P-gp inhibitors have appeared today [122]. Although the second- and third-generation P-gp inhibitors have higher affinity and lower side effects, their therapeutic effects have not been improved as expected in preclinical and clinical studies. Therefore, it will be exciting to develop an effective, high-affinity and relatively non-toxic natural substance as an inhibitor. P-gp modulators derived from natural products belong to the fourth generation of P-gp inhibitors [123]. A variety of studies have reported natural products as P-gp inhibitors. For example, cryptotanshinone and dihydrotanshinone, the extracts of danshen, can reverse the sensitivity to doxorubicin and irinotecan by downregulating the expression of P-gp mRNA and protein in CRC cell lines [12]. Natural products cinobufagin **[13]** and 8-oxocoptisine [14] have also been shown to reverse chemotherapy resistance for CRC by inhibiting P-gp. This evidence indicates that natural products may be a target for solving chemotherapy resistance in the future. In addition to P-gp inhibitors, other modulators of ABC transporters have achieved good results. Most of the experimental data about ABC transporter inhibitors are obtained based on cell experiments, and the specific effects in vivo have not been well-proven. Therefore, the main work included preclinical or clinical research based on pharmacology.

Another way to inhibit the ABC transporter is to reduce the expression level of the transporter. Studies have shown that antisense oligonucleotides (AOSs), miRNA and LncRNA can mediate the expression of ABC transporters to reverse drug resistance. For example, liposome AOSs reverse multi-drug resistance in vivo and in vitro by inhibiting the expression of P-GP, MRP1, MRP2 and Bcl-2/Bcl-XL, which may enhance the sensitivity of CRC to chemotherapy by decreasing the activity level of the MDR1 promoter [15,16]. MiR-506 and miR-26b are downregulated in chemotherapy-resistant CRC cell lines and specimens. Overexpression of miR-506 and miR-26b can reverse drug resistance of CRC cells to oxaliplatin and 5-FU by downregulating P-gp [117,124]. Recent studies have shown that LncRNA CASC15 is highly expressed in oxaliplatin-resistant CRC cell lines and specimens. LncRNA CASC15 can enhance the expression of ABCC1 by inhibiting the expression of miR-145 from conferring oxaliplatin resistance to CRC in vivo and in vitro [125]. LncRNA PVT1 was overexpressed in CRC cells and specimens with 5-FU resistance, which might be attributed to the fact that PVT1 increased CRC resistance to 5-FU by inhibiting apoptosis and upregulating the expression of MRP1, P-GP and Bcl-2 mRNA and protein. Silencing LncRNA PVT1 increased apoptosis and downregulated ABC transporter, thus increasing the sensitivity of CRC to 5-FU [126]. Therefore, it is necessary to develop strategies based on antisense oligonucleotides, miRNA and LncRNA to reduce the expression level of ABC transporters to reverse the drug resistance of CRC cells.

## 4. Changes of Drug Targets

Aberration of corresponding drug targets and/or ligands can mediate the generation of drug resistance, no matter whether for chemotherapy drugs or molecular targeted therapy drugs (Figure 2).

### 4.1. Thymidylate Synthase (TS)

Changes in target mutations or expression levels of chemotherapeutic drugs can influence drug responses and subsequently induce drug resistance. 5-FU mainly destroys the deoxynucleotide pool required for DNA replication by inhibiting thymidylate synthase (TS), and mixes its metabolites into RNA and DNA to exert anti-tumor effects [4,127]. Several studies have shown that the high expression of TS is a crucial determinant of CRC resistance to 5-FU, and 5-FU intervention can also increase the expression of TS [4,9,10]. Under normal circumstances, unbound TS can be combined with its own mRNA, thus inhibiting its translation through the activity of decreased negative feedback, but when TS with 5-FU metabolites FdU experiences steady binding, TS can no longer bind to its own mRNA, resulting in increased protein expression because the sharp increase in TS will promote the recovery of enzyme activity, which constitutes a potential resistance mechanism [128]. Therefore, developed methods to inhibit this adverse feedback pathway will be a new way to combat resistance. In addition, the addition of folic acid sources such as leucovorin to exogenous therapy enhances the cytotoxicity of 5-FU by increasing TS inhibition in vitro and in vivo [129].

### 4.2. Topoisomerase 1 (TOP-1)

Given the targeting properties of irinotecan or its active metabolite SN-38, mutations or lower expression of TOP-1 may cause irinotecan resistance to CRC [89]. The rearrangement and hypermethylation of TOP-1 genes can be found in drug-resistant cells, leading to transcriptional silencing [130]. In addition, the amount of TOP-1 bound to DNA is reduced, which may also lead to CRC resistance to irinotecan [88,131]. Therefore, the increased expression of TOP-1 may increase irinotecan sensitivity. Y-box binding protein 1 (YB-1), as a promoter of intracellular TOP-1 catalytic activity, can improve the sensitivity of irinotecan by directly interacting with TOP-1 [17].

### 4.3. Epidermal Growth Factor Receptor and Its Ligand

Cetuximab specifically binds epidermal growth factor receptor (EGFR) and blocks the EGFR signaling pathway, thereby exerting anti-tumor activity [132]. However, multiple ligands of EGFR can activate EGFR, such as epidermal growth factor (EGF), transforming growth factor alpha (TGF-α), heparin-binding-EGF (HB-EGF), betacellulin (BTC), amphiregulin (AREG) and epiregulin (EREG) (Figure 2) [133,134]. Several clinical studies have confirmed that the expression levels of AREG and EREG are related to the treatment response of cetuximab [135,136]. However, these ligands have no predictive power in KRAS-mutations patients [136]. Similar results were obtained in a subsequent randomized clinical trial of panitumumab, irinotecan and cyclosporine to treat colorectal cancer [137]. This evidence suggests that upregulation of AREG and EREG may be one of the mechanisms of anti-EGFR mAbs resistance in KRAS-WT (wild-type) CRC patients.

Mutations or low expression of EGFR are detrimental to the efficacy of anti-EGFR therapy in lung cancer [138]. However, clinical studies have shown that metastatic colorectal cancer (mCRC) patients respond to anti-EGFR mAbs regardless of EGFR status [139,140]. A growing number of studies suggest that mutations in the extracellular domain (ECD) of the EGFR, such as S492R, G465E, R451C and K467T, prevent binding cetuximab and make CRC resistant to anti-EGFR mAbs [141,142,143,144]. In addition, methylation at R198 and R200 of the EGFR ECD plays an essential role in regulating EGFR functionality and resistance to cetuximab treatment in CRC patients [145].

Therefore, the development of new mAbs that can bind to different or mutated EGFR ECDs is expected to improve the efficiency of anti-EGFR mAbs. MM-151 is a drug consisting of three anti-EGFR mAbs. Preclinical studies have shown that CRC with resistance to cetuximab or panitumumab due to EGFR ECD mutations is sensitive to EGFR blockade by MM-151 [19]. A Phase I clinical trial reported that MM-151 has an acceptable tolerability profile and objective clinical activity alone and in combination with irinotecan [64]. Necitumumab is another FDA-approved anti-EGFR mAbs. A preclinical study has shown that necitumumab can bind and inhibit the most common mutated form of EGFR (S468R) that is resistant to cetuximab and other EGFR epitope variants that are resistant to cetuximab [20]. A Phase II trial study evaluated necitumumab plus modified FOLFOX6 as a first-line treatment for locally advanced or metastatic CRC [65]. Compared with unselected mCRC patients who received mFOLFOX6 alone in a randomized Phase III TREE study, the addition of necitumumab significantly improved objective response rate (ORR, 63.6% vs. 41%) and overall survival (OS, 22.5 m vs. 19.2 m) [65,146]. Sym004 is a 1:1 mixture of two recombinant human-mouse chimeric mAbs directed against non-overlapping EGFR epitopes. Preclinical studies have shown that Sym004 could effectively inhibit cells stimulated by EGFR ligands and overcome cetuximab resistance mediated by EGFR ECD mutations in CRC [18,147]. However, in a Phase II clinical study, Sym004 effectively targeted cancer cells with EGFR ECD mutations, but did not improve the OS of mCRC [63]. Nevertheless, EGFR-related testing in mCRC is not currently recommended [148].

## 5. Aberration in Downstream Signaling Pathways

The EGFR signaling pathway has two downstream pathways: RAS/RAF/MEK/ERK and PI3K/AKT/mTOR. Anti-EGFR mAbs, such as cetuximab, inhibit these downstream pathways by inhibiting EGFR, so aberrant activation of these downstream pathways can mediate anti-EGFR mAbs resistance by bypassing EGFR (Figure 2).

### 5.1. RAS/RAF/MEK/ERK

RAS mutations are well-known predictors of resistance to anti-EGFR mAbs [135]. Some large clinical trials have shown that FOLFIRI (leucovorin + 5-FU + irinotecan) or FOLFOX regimens combined with EGFR inhibitors can benefit KRAS-WT CRC. However, CRC with KRAS mutations do not benefit from EGFR inhibitors, possibly because activation of KRAS does not depend upstream on EGFR [149,150,151,152]. KRAS mutations are the most common RAS mutations, occurring in 40% of mCRC [149]. Mutations in exons 2–4 of KRAS and NRAS can lead to mCRC resistance to cetuximab and panitumumab [153,154]. However, there are still no drugs that specifically target RAS. Some preclinical and clinical studies have reported that some drugs could reverse resistance induced by RAS, such as sotorasib, dasatinib and metformin [155,156,157].

BRAF mutations are another critical factor in the poor prognosis of mCRC patients treated with anti-EGFR mAbs [149,152]. The hotspot of BRAF mutations in CRC is the substitution from valine to glutamic acid at codon 600 (V600E), located in exon 15, leading to 130- to 700-fold increased BRAF kinase activity compared with that of BRAF-WT [158]. The prevalence of BRAF V600E mutations in mCRC is 8–10%, and they occur only in KRAS-WT mCRC [159,160]. Vemurafenib is an oral inhibitor that binds explicitly to BRAF V600 kinase. A pilot trial in patients with BRAF-mutations mCRC after chemotherapy showed that the combination of vemurafenib and panitumumab limited tumor progression and produced moderate clinical activity [66]. However, in another clinical study, CRC patients did not benefit from vemurafenib monotherapy and/or cetuximab [161]. In a Phase Ib trial study, vemurafenib combined with irinotecan and cetuximab had favorable survival outcomes and response rates in patients with refractory BRAF-mutant mCRC [67]. Encorafenib, another BRAF inhibitor, has been shown to reverse anti-EGFR resistance by targeting both EGFR and BRAF. In the recently published BEACON CRC study, encorafenib plus cetuximab improved OS, ORR and progression-free survival in 444 patients with previously treated BRAF V600E-mutant mCRC compared with standard chemotherapy [162].

MEK/ERK is a downstream cascade of RAS/RAF. We should emphasize that RAS/RAF mutations ultimately lead to cell proliferation and survival by activating MEK. It is exciting that multiple preclinical studies have demonstrated that MEK inhibitors can inhibit the proliferation of CRC cells with KRAS or BRAF mutations in vitro and in vivo, reversing drug resistance mediated by KRAS or BRAF mutations [21,22]. However, KRAS or BRAF mutant CRC usually develops resistance to MEK inhibitors [25,163]. Although the mechanism of this resistance is unclear, preclinical studies suggest that it may be related to the reactivation of the ERK signaling pathway [25]. SCH772984, a novel ERK inhibitor, effectively inhibited MAPK signaling and cell proliferation in BRAF or MEK inhibitor resistance models and reversed the sensitivity of including CRC to BRAF and MEK inhibitors [25].

### 5.2. PI3K/AKT/mTOR

PI3K/AKT/mTOR is another downstream parallel pathway of EGFR. There is increasing evidence that aberrant activation of this pathway is associated with anti-EGFR treatment resistance [164]. In a clinical study of 110 mCRC patients treated with anti-EGFR mAbs, the authors found that PIK3CA mutations were significantly associated with clinical resistance to panitumumab or cetuximab, particularly in KRAS wild-type mCRC [165]. PIK3CA mutations occurring in the “hotspots” located in exon 9 (E542K, E545K) and exon 20 (H1047R) are oncogenic in CRC cellular models [166]. However, PIK3CA exon 20 mutations significantly reduced the anti-EGFR response rate in the case of KRAS-WT, while exon 9 mutations had no significant effect [167,168]. In addition, tumor suppressor gene phosphatase and tensin homolog (PTEN) is a negative regulator of this pathway and aberrant PTEN may lead to anti-EGFR resistance [169]. Preclinical and clinical data show that loss of PTEN protein significantly reduces the sensitivity of CRC to cetuximab [170,171].

In summary, these changes confer CRC resistance to anti-EGFR mAbs. Whereas PI3K inhibitors and mTOR inhibitors have been developed and approved for clinical trials in PIK3CA-mutated solid tumors [164]. However, two clinical trials showed that adding PI3K inhibitors to cetuximab did not improve PFS, ORR and OS in mCRC patients [68,69]. More and more clinical trials are needed to verify its safety and efficacy.

## 6. Aberrant Activations of Alternative Receptors

Insulin-like growth factor 1 (IGF1)/IGF2, HGF/MET and HER2/HER3 have been reported to activate the PI3K/AKT and RAS/RAF/MEK/ERK axis independently of EGFR [172,173]. Alterations in these pathways play an essential role in primary and secondary resistance to anti-EGFR mAbs (Figure 2).

### 6.1. IGF1 and IGF2

IGF1 and IGF2 have substantial effects on cell proliferation and differentiation and are effective inhibitors of apoptosis [174]. The role of IGF1 and IGF2 is mediated mainly by the IGF1 receptor (IGF-1R). Multiple clinical studies have demonstrated that IGF1-negative patients with CRC in KRAS-WT treated with irinotecan-cetuximab have higher responses, higher median-PFS and higher median-OS than IGF1-positive patients [172,175]. In addition, preclinical studies have shown that IGF2 overexpression attenuates the efficacy of cetuximab in CRC patient-derived xenografts (PDXs) [176].

Targeting IGF1/IGF2 may be an effective way to overcome anti-EGFR resistance. In preclinical studies, co-targeting EGFR and IGF-1R can synergistically enhance the anti-tumor effect of CRC cells [26]. Unfortunately, IGF-1R inhibitor IMC-A12 [70], dalotuzumab (MK-0646) [71] and ganitumab [72] did not induce effective ORR and improved median-OS in mCRC. This is inconsistent with the theoretical efficacy of IGF-1R inhibitors, and more clinical evidence and new targeted drugs are needed to demonstrate the possibility of IGF-1R as a therapeutic target after anti-EGFR mAbs resistance.

### 6.2. HER2 and HER3

The cancer genome atlas (TCGA) colorectal cancer project found that 7% of CRC patients have HER2 somatic mutations or HER2 gene amplification [177]. A series of studies have demonstrated that HER2 gene amplification or activating mutations can mediate resistance to anti-EGFR mAbs in CRC [27,178,179]. It may be that HER2 gene amplification or activating mutations mediate aberrant activation of HER2, leading to persistent ERK1/2 signaling mediated resistance to cetuximab [178,179].

Therefore, targeting HER2 may be a strategy to reverse CRC resistance to EGFR. Current HER2-targeted drugs, such as trastuzumab, pertuzumab and lapatinib, have significantly improved HER2-amplified breast cancer and CRC. Preclinical studies have shown that the combination of HER2 and EGFR blockade can lead to CRC tumor regression [27,28,29]. In addition, in CRC, the HER2-mutated PDXs model showed that trastuzumab plus neratinib caused sustained tumor regression compared to a single HER2-targeted agent [179]. In two large clinical studies, trastuzumab and lapatinib/pertuzumab combinations showed good activity and tolerability in EGFR-resistant HER2-positive mCRC [74,75]. However, data from a recently published Phase Ib clinical study suggest that neratinib-plus-cetuximab did not show an objective response in quadruple-WT (KRAS, NRAS, BRAF and PIK3CA) mCRC resistant to cetuximab or panitumumab [73].

HER3 mutations are present in 5.7% to 11% of CRC [180,181]. Extensive data suggest that aberrant HER3 in mCRC may also reduce the efficacy of anti-EGFR therapy [172,182]. Currently, HER3-targeted drugs U3-1402 [30], lumretuzumab [76] and MEHD7945A [77] have been used in preclinical and clinical trials with good safety and efficacy. Further clinical studies are needed to demonstrate the clinical value of HER2/HER3 combined with anti-EGFR mAbs.

### 6.3. HGF/MET

The proto-oncogene MET, which enforces hepatocyte growth factor receptor (HGFR, also known as MET tyrosine kinase receptor), shares several downstream pathways with EGFR, including RAS/RAF/MAPK and PI3K/AKT signaling pathways [173,183]. A clinical study has demonstrated that MET amplification is associated with acquired resistance to cetuximab or panitumumab in patients with KRAS-WT mCRC [184]. In addition, anti-EGFR therapy leads to MET amplification, contributing to the development of CRC-acquired resistance [185,186]. The possible reason is that EGFR ligand TGF-α can induce cross-interaction between EGFR and MET, which increases the phosphorylation of MET and its downstream MAPK and AKT, leading to the generation of drug resistance [187].

Therefore, targeting HGF-MET may be a way to overcome resistance to anti-EGFR therapy. Tivantinib is a selective oral MET inhibitor that has been used in monotherapy and combination therapy for solid tumors, including mCRC [188]. Data from a Phase II clinical study showed that tivantinib combined with cetuximab showed good toleration and activity in 41 mCRC patients with MET-high who relapsed after cetuximab or panitumumab treatment [78]. However, in another clinical study, the combination of tivantinib, cetuximab and irinotecan was well-tolerated in patients with KRAS-WT mCRC, but did not significantly improve PFS [79]. In addition, rilotumumab (AMG 102) is an all-human HGF-targeting IgG 2 mAbs that neutralizes HGF-dependent MET signaling. In a randomized Phase Ib/II trial, the combination of panitumumab and rilotumumab significantly increased the anti-tumor activity of patients compared with panitumumab alone [72].

## 7. Persistent Activation of Oncogenic/Bypass Signaling

### 7.1. Wnt/β-Catenin

As an evolutionarily conserved signal, Wnt/β-catenin signaling is involved in various processes of embryonic development and also plays a crucial role in cancer [189]. Sustained activation of Wnt/β-catenin signaling endows cancer cells with sustained self-renewal growth and is associated with tumor treatment resistance [190].

Current studies have shown that activation of the Wnt/β-catenin signaling pathway can mediate CRC resistance to chemotherapy through several pathways, such as upregulation of MDR1, inhibition of apoptosis, maintenance of stem cell dryness, regulation of epithelial-mesenchymal transformation (EMT) and regulation of tumor microenvironment [191]. In addition, Wnt and EGFR signals are closely related. For example, EGFR can form a complex with β-catenin to activate the Wnt pathway, and the Wnt ligand can confer anti-EGFR resistance through its Frizzled activation of EGFR signaling [192,193]. The overexpression of MiR100HG and miR-100/125b was also observed in cetuximab-resistant CRC cell lines and tumors from CRC patients that progressed on cetuximab [194]. Specifically, miR-100/125b coordinately represses five Wnt/β-catenin negative regulators, resulting in increased Wnt signaling, and inhibition of Wnt restored cetuximab reactivity in cetuximab-resistant cells [194].

Immune checkpoint inhibitors (ICIs) such as nivolumab and pembrolizumab could reverse T cell dysfunction and apoptosis by inhibiting programmed death-1 (PD-1), which enhances T cell activation and cytotoxicity to tumor cells [195]. The Wnt/β-catenin signaling pathway is also associated with ICIs resistance. In primary CRC tumor samples, Wnt signaling reduced the frequency of tumor-infiltrating T cells, regardless of mutation load [196]. In addition, APC, as a gene that upregulates WNT signaling, is negatively correlated with the frequency of T cell infiltration [196,197,198]. It may be that the loss of APC upregulates the expression of Dickkopf-related protein 2 (DKK2), which, together with its receptor LRP5, inhibits STAT5 signaling, leading to immunosuppression. Induction of DKK2 ablation activates natural killer cells (NK) and CD8+ T cells in tumors and enhances the efficacy of anti-PD-1 therapy [198].

Inhibition of the Wnt pathway may reverse CRC sensitivity to treatment. For example, inhibiting the expression of transient receptor potential channel 5 (TRPC5) can weaken the ABCB1 efferent pump by inhibiting the Wnt/β-catenin signaling pathway, thus reversing 5-FU resistance in CRC [31]. The silencing of aquaporin 5 (AQP5) enhances CRC sensitivity to 5-FU by inhibiting the Wnt/β-catenin signaling pathway [32]. Blocking of Wnt/β-catenin signaling improves CD8 + T cell infiltration and initiation, creating a more favorable situation for immune checkpoint inhibition [199]. Combination therapy with Wnt inhibitors and immune checkpoint inhibition is an exciting strategy that should be evaluated in preclinical and clinical studies. However, active Wnt signaling disrupts the transcriptional activity of Foxp3, which is critical for the development and function of regulatory T cells (Tregs) [200]. Therefore, it must be recognized that blocking Wnt for immune regulation is still controversial at present. In conclusion, the Wnt/β-catenin signaling pathway may be a promising target for reversing tumor resistance.

### 7.2. JAK/STAT

The Janus kinases (JAK) and signal transducer and activator of transcription (STAT) are essential in regulating cell survival, proliferation, differentiation and apoptosis as the convergence points of many oncogenic signaling pathways [201]. STAT3, a member of the STAT transcription factor family, is upregulated in various tumors, including CRC [202]. It has been shown that STAT3 expression level in CRC is positively correlated with chemoradiotherapy resistance [203]. This may be due to IL-6-induced STAT3 phosphorylation, and inhibition of STAT3 reverses CRC resistance to chemotherapy in vivo and vitro [203]. In addition, nuclear pyruvate kinase isoform M2 (PKM2) promotes gefitinib resistance by upregulation of STAT3 activation in CRC, and inhibition of STAT3 or PKM2 can reverse CRC against EGFR resistance [204].

IFN-γ is mainly produced by T cells, NK cells and NK T cells, and IFN-γ signal must be activated through JAK1/2/STAT1 pathway, suggesting that IFN-γ/JAK1/2/STAT1 pathway may be involved in anti-tumor immune response [205,206]. Initially, IFN-γ signaling can play an anti-tumor role by increasing antigen presentation, such as upregulation of major histocompatibility complex (MHC) molecules and recruitment of immune cells [207]. However, continued IFN-γ activation causes immune reprogramming of tumor cells, leading to tumor immune escape [208]. Tumor cells evade IFN-γ by downregulating or mutating the molecules involved in the IFN-γ signaling pathway, such as JAK1/2 and STAT, thus blocking the IFN-γ signaling pathway. Preclinical studies have shown that defects in IFNGR1 (IFN-γ receptors) and JAK1/2 genes cause tumor cells to be unresponsive to IFN- γ, which allows tumor cells to grow in the presence of IFN- γ [209]. JAK1/2-inactivating mutations reduce programmed death ligand 1 (PD-L1) and MHC I expression in mouse and human CRC [33,210]. This implies that the inactivation mutations of JAK1 and JAK2 block the IFN-γ signaling pathway, leading to downregulation of tumor PD-L1 expression, so it would be unnecessary to attempt anti-PD-L1 therapy in this case. Inactivation mutations of JAK1 and JAK2 have been observed in a biopsy specimen of high TMB colon cancer that did not respond to PD-1 therapy, further highlighting the importance of IFN-γ deficiency against PD-1 resistance [210]. In addition, the expression of MHC class I molecules in a variety of tumors mainly depends on IFN-γ signal [205]. Preclinical studies have shown that hypoxia induces a decrease in IFN-γ signaling in a mouse CRC model, leading to a decrease in MHC class I molecule expression [211]. A positive correlation between IRF2 expression and MHC I has been observed in many human tumors, including CRC [212].

Multiple pathways have been developed, such as toll-like receptor (TLR) agonists, oncolytic viruses or other pathways that activate alternative interferon pathways (type I IFN) that lead to signal transduction and transcriptional activator 1 (STAT 1) and STAT 2 signaling, thereby promoting transcription of PD-L1 and MHC classes by inducing IRF1 [33,213].

### 7.3. TGF-β

TGF-β signaling activation plays a key role in cancer progression and is associated with tumor treatment resistance [214]. Preclinical studies have shown that 5-FU can activate the TGF-βRI/Smad3 pathway to confer chemotherapeutic resistance to CRC; TGF-βRI inhibition reduces the proliferation of chemoresistant cancer cells and increases cell death [215]. In addition, it has been reported that the effect of TGF-β on chemotherapy resistance may require the coordination of protease-activated receptor 2 (PAR2) because the activation of PAR2 eliminates the inhibition of TGF-β [216]. In addition, TGF-β receptor maturation may be disrupted by mediator protein complex subunit 12 (MED12), which leads to activation of TGF-β signaling pathway and upregulation of mesenchymal marker expression, further conferring chemotherapeutic tolerance in CRC patients [217]. MED12 deletion, TGF-β receptor overexpression or recombinant TGF-β therapy induces TKI resistance in various solid tumors, including CRC [37]. This may be due to TGF-β signaling inducing MEK/ERK signal activation, thus restoring the reduction in TKI-mediated MAPK pathway inhibition. Therefore, MED12 may be a promising target in the future.

TGF-β can provide an escape route for tumor angiogenesis by influencing and regulating tumor angiogenesis [218]. Preclinical studies have shown that TGF-β receptor I inhibitors (SD-208) inhibit anti-angiogenesis in hepatocellular carcinoma (HCC) and glioblastoma multiforme (GBM) xenograft tumors [219,220]. However, SD-208 did not cause significant vascular inhibition in the CRC model, at least in the SW-48 cell line [221]. This suggests that TGF-β-mediated angiogenesis may be related to the corresponding tumor microenvironment. Activin receptor-like kinase 1 (ALK-1) is a TGF-β transmembrane receptor that plays a crucial role in TGF-β -mediated angiogenesis [218]. PF-03446962 is an IgG2 monoclonal antibody against human ALK-1. PF-03446962 has been shown in preclinical studies to reverse the sensitivity of tumors to antiangiogenic therapy [38]. Regorafenib is a multikinase inhibitor that targets VEGF R1-3, KIT and PDGFR-α/β. However, in a Phase Ib trial study, regorafenib combined with PF-03446962 did not show significant clinical activity in patients with refractory mCRC [81].

TGF-β signaling can produce profound immunosuppressive activity on crucial cell types of innate and adaptive immunity, thereby attenuating the inherent anti-tumor potential of immune cells within TME [214]. For example, mouse CRC models with low mutation, T cell rejection and TGF-β activation have difficulty benefiting from immune checkpoint inhibitors. This may be due to increased TGF-β promoting T cell rejection and blocking the Th1 effector phenotype. Blockade of the TGFβ signaling pathway promotes enough T cell infiltration to make tumors sensitive to PD-1 therapy [39]. Similar therapeutic responses to combined TGF-β and PD-L1 blockade were observed in other mouse CRC models [222,223].

### 7.4. EGFR

#### 7.4.1. PI3K/AKT/mTOR

The activation of PI3K/AKT/mTOR signaling pathway would occur in 20% of CRC patients, and the activation of this pathway may be related to the expression of PD-L1 [224,225]. A series of preclinical studies has shown that PI3K/AKT induces high expression of PD-L1 in colon cancer stem cells and mediates immune escape [226,227]. In addition, as a negative regulator of PI3K/Akt/mTOR signaling pathway, loss of PTEN leads to upregulation of PD-L1 protein, decrease of CD8 + T cell and increase of TAM, which are negative predictors of PD-1 blocking response in microsatellite instability-high (MSI-H) CRC patients [228,229]. Inhibition of the PI3K/AKT/mTOR pathway improves the efficacy of anti-PD-1 and anti-CTLA-4 antibodies in mouse models [226,230,231].

#### 7.4.2. RAS/RAF/MEK/ERK

The increased expression of PD-L1 has been observed in the specimens and cell lines of CRC patients with KRAS mutation, which may increase the stability of PD-L1 mRNA by regulating the AU-rich element-binding protein tristetraprolin (TTP), and the combined blocking of anti-EGFR and PD-1/PD-L1 significantly inhibits tumor growth [232,233]. In addition, recently published preclinical data suggest that KRAS mutations enhance resistance to PARPi and anti-PD-L1 therapy in CRC PDX models, which may be related to PMEK1/2 and pMAPK, and the addition of MEKi to the combination of PARPi and anti-PD-L1 therapy reverses drug resistance in KRAS-mutated tumors [23].

The MAPK axis is the main downstream pathway triggered by normal T cell receptors (TCRs) [234]. Continuous TCR stimulation drives negative regulator PD-1 expression and inhibits effector T cell function [235]. MEK is an essential part of MAPK signaling pathway, and inhibition of MEK may rejuvenate anti-tumor T cells [236]. In a mouse CRC model, MEK inhibition protects CD8+T cells from chronic TCR-stimulus-driven death while preserving cytotoxic activity. Combining MEK inhibition with anti-PD-L1 produces collaborative and durable tumor regression [24]. Moreover, MEK inhibits apoptosis-inducing markers, upregulates HLA expression and downregulates immunosuppressive factors such as PD-L1, IL1, IL8, NT5E and VEGFA [237]. This evidence suggests that immunoregulation by MEK inhibitors combined with immune checkpoint inhibitors is a more effective anti-tumor therapy. A Phase Ib trial study evaluated atezolizumab plus cobimetinib in chemical-refractory mCRC. This combination therapy had manageable safety and clinical activity [82]. However, in another Phase III trial study, the atezolizumab plus cobimetinib did not increase the tumor response or survival benefit of mCRC compared to treatment alone [83]. Moreover, a deleterious effect of MEK inhibition on T lymphocyte proliferation and antigen-specific T lymphocyte activation has been observed in melanoma [238,239]. This may partly explain why CRC patients do not benefit from MEK inhibitors and immune checkpoint inhibitors.

## 8. Pathway of Cell Death

When enough active drugs accumulate and inhibit their corresponding targets, treatment outcomes depend on how the cancer cells respond. Ideally, drug-induced damage is closely associated with cell death. Apoptosis, autophagy and ferroptosis are the main modes of cell death pathway [240]. The dysfunction of programmed death pathways is related to tumor drug resistance (Figure 2).

### 8.1. Apoptosis

Intrinsic or acquired resistance to apoptosis is one of the characteristics of human cancers [241]. Apoptosis can pass through two signal transduction pathways, such as the death receptor (external) pathway and the mitochondrial (internal) pathway, both of which ultimately lead to the cleavage of the executioner caspase-3 [241]. Therefore, aberration of these pathways may amplify the degree of tumor treatment resistance by inhibiting apoptosis pathways.

The internal apoptosis pathway is mainly regulated by members of the Bcl-2 family [242]. The Bcl-2 protein family consists of anti-apoptotic proteins, such as Bcl-2, Bcl-XL and McL-1, and pro-apoptotic molecules, such as Bax and BH3-only proteins [243]. Dysregulation of antiapoptotic protein and proapoptotic protein may induce drug resistance in the tumor. Preclinical studies have shown that 5-FU leakage will increase the expression of Bcl-2 and Bax in CRC, and the decrease or deletion of Bax expression will increase the resistance of CRC to 5-FU and OXA [244].

Therefore, targeting the Bcl-2 protein family may be an effective way to reverse drug resistance. ABT-737 is a small molecule BH3 analog. ABT-737 and its oral derivative ABT-263 (navitoclax) can target anti-apoptotic proteins and promote the pro-apoptotic ability of Bax and Bak [245]. ABT-263 in combination with selumetinib increased caspase-dependent cell death through Bax and effectively reduced the incidence of CRC-acquired resistance to selumetinib [40]. However, resistance mechanisms limit the effectiveness of these drugs. The most apparent cause is another anti-apoptotic member of the Bcl-2 protein family, McL-1 [245,246]. Obatoclax is a promising novel agent that simultaneously targets members of the entire anti-apoptotic Bcl-2 protein family, including McL-1 [247]. However, there are few studies on CRC by obatoclax. Timme et al. [248] showed that obatoclax could restore oxaliplatin-induced CRC cell apoptosis by blocking McL-1 and/or Bcl-XL.

The inhibitor of apoptosis protein (IAP) family is a crucial regulator of programmed death pathway, which can block the activation of caspase and negatively regulate apoptosis pathway [249]. The IAP family mainly includes neuronal IAP (NIAP), cellular IAP1 (cIAP1), cellular IAP2 (cIAP2), X chromosome link IAP (XIAP) and survivalin [250]. Preclinical studies have shown that XIAP can promote resistance of CRC cells to apoptosis induced by multiple tumor treatments [251,252,253]. miR-587 enhances AKT activation and XIAP signal transduction by downregulating the expression of negative AKT regulator PPP2R1B, thereby inhibiting 5-FU-induced apoptosis [254].

AEG35156 is a new second-generation antisense oligonucleotide targeting XIAP. In the mice CRC model, AEG35156 can reduce XIAP mRNA and significantly enhance the therapeutic effect of taxanes [41]. Second mitochondria-derived activator of caspases (SMAC) can inhibit IAP function, thereby inhibiting cell apoptosis. In the past 20 years, several small molecules simulating SMAC function have been developed [255]. Some studies have confirmed that birinapant, a SMAC analog, can improve the response of CRC cells to oxaliplatin/5-FU [42]. XIAP inhibitors and SMAC activators are promising drugs that enhance anti-cancer drug-mediated apoptosis of CRC cell.

DNA damage caused by chemotherapeutic drugs always leads to apoptosis of CRC cells. As a tumor suppressor gene, P53 mutation could disrupt DNA-damage-induced cell cycle arrest and inhibit tumor cell apoptosis, which could induce chemotherapy resistance in various solid tumors, including CRC [256,257]. A series of clinical studies have confirmed that CRC patients with wild-type P53 can benefit from 5-FU, and CRC patients with mutant P53 have a poor prognosis when receiving adjuvant chemotherapy [258,259,260].

Enhancing the activity of P53 may be a way to reverse chemotherapy resistance in CRC. Murine double minute 2 (MDM2) is a primary physiological antagonist of P53, which is inhibited by hiding the activation domain of P53 or by ubiquitin-ligase [261,262]. Therefore, the development of MDM2-based inhibitors may help reverse drug resistance. Nutlin-3 is a specific antagonist of MDM2-p53 interaction. Increased p53 activity has been reported after nutlin-3 treatment in breast, prostate and CRC cells [263,264]. In addition, TRIM67 interacted directly with the C-terminus of p53, inhibiting p53 degradation by its ubiquitin ligase MDM2 [265]. Preclinical studies have shown that TRIM67 can enhance the anti-proliferation effect of adriamycin, 5-FU or oxaliplatin in p53 wild-type CRC cells [265].

### 8.2. Autophagy

Autophagy is a self-degradative process that is important for balancing energy sources at critical times in development and in response to nutrient stress [266]. The role of autophagy in tumors is controversial and highly dependent on tumor stage and classification [267]. Autophagy acts as a tumor suppressor in precancerous lesions and inhibits tumorigenesis through type II programmed cell death [268]. Autophagy acts as a tumor promoter in identified tumors, which can degrade damaged proteins and organelles as reserve energy to supply tumor cells for survival in anti-cancer therapy and endue tumor cells with resistance to tumor therapy [269]. In the process of autophagy, different autophagy-related genes (ATG) are always regulated, such as ATG1 (ULK1), Vps34 (PIK3C3), Beclin1 (BECN1), LC3 and ATG5 [267].

In CRC patients undergoing chemotherapy, overexpression of autophagy-associated proteins, such as Beclin-1, was associated with low survival [270]. These results suggest that autophagy-associated proteins may be associated with chemotherapy resistance in CRC patients. Also, abundant *Clostridium* nuclei have been detected in CRC specimens of patients, with recurrence after chemotherapy. Clostridium nuclei activate autophagy pathways and alter the chemotherapy response of CRC by targeting TLR4 and MYD88 innate immune signaling and specific microRNA [271]. IL-6 activates autophagy and promotes chemotherapy resistance in CRC through the IL-6/JAK2/BECN1 pathway [272]. Mechanistically, IL-6 promotes the phosphorylation of BECN1-Y333 through JAK2 signaling, leading to autophagy [272]. In addition, non-coding RNA can mediate CRC resistance to chemotherapy by enhancing autophagy pathways, such as circHIPK3, LncRNA SNHG6, LncRNA SNHG14, LncRNA H19 and LncRNA. NEAT1 has been shown to mediate CRC resistance to chemotherapy by enhancing the autophagy pathway [273,274,275,276,277]. A series of preclinical studies have demonstrated that autophagy inhibitors, such as chloroquine/3-methyladenine, can reverse CRC resistance by inhibiting 5-FU-induced activation of autophagy [43,44].

Elevated levels of autophagy-associated protein have also been observed in patients with advanced CRC treated with cetuximab, and autophagy-associated protein levels predict patient response to cetuximab [278]. It is possible that cetuximab induces autophagy in cancer cells by downregulating HIF-1α and Bcl-2 and activating the Beclin1/hVps34 complex. Autophagy inhibitor, such as chloroquine/3-MA, could enhance the sensitivity of CRC cells to cetuximab-induced cell death [45,46].

There is increasing evidence that autophagy is involved in immune system development and the survival and function of effector T cells [279]. Therefore, autophagy may be involved in tumor immune escape, which is based on immunogenicity loss. For example, in a mouse CT26 colon tumor model, mitoxantrone and oxaliplatin activate autophagy and enhance tumor-infiltrating dendritic cells and T cells, and depletion of autophagy-associated proteins (ATG5 or ATG7) reduces chemotherapy-induced anti-tumor immune responses [280]. In a mouse CRC model, the autophagy inhibitors, such as SB02024 and SAR405, increased NK and T cell infiltration in TME and reversed resistance to anti-PD-1 or anti-PD-L1 [47]. A clinical trial evaluates the combination of the autophagy inhibitor hydroxychloroquine, immune checkpoint inhibitor atezolizumab and MEK inhibitor cobimetinib in patients with advanced gastrointestinal tumors with KRAS mutations (NCT04214418).

### 8.3. Ferroptosis

Unlike autophagy and apoptosis, ferroptosis is a type of cell death caused by iron-dependent and lipid reactive oxygen species (ROS), first proposed by Dixon and colleagues in 2012 [281]. Several drugs, such as cisplatin and cetuximab, could induce ferroptosis, thus exerting its anti-tumor effect, which involves several vital molecules, NOX and P53, glutathione peroxidase 4 (GPX4), nuclear factor E2-related factor 2 (Nrf2) and solute carrier (SLC) family members. For example, GPX4, as a negative regulator of iron apoptosis, plays a major antioxidant role in anti-cancer activity and the inhibition of GPX4 can enhance ferroptosis and thus enhance the sensitivity of CRC to cisplatin/oxaliplatin [49,282]. Lipocalin 2 is an iron carrier binding protein that can regulate ferroptosis. It has been proven that lipocalin 2 inhibits ferroptosis and leads to 5-FU resistance in colon cancer cell lines [283]. The transcriptional regulator Nrf2 is another crucial inhibitor of iron apoptosis because it inhibits iron uptake, restricts ROS accumulation and upregulates SLC7A11 [284]. Higher Nrf2 was observed in CRC cells with 5-FU resistance, which further led to increased heme oxygenase-1 (HO-1) expression and reduced CRC sensitivity to 5-FU [285].

As mentioned earlier, RAS mutations limit the usefulness of anti-EGFR in mCRC patients. Ras-selective lethal 3 (RSL3) is a small molecule that kills RAS-mutated cancer cells and activates iron apoptosis in RAS-mutated cancer cells [286]. In KRAS mutant CRC cell lines and mouse CRC models, cetuximab promotes RSL3-induced ferroptosis by activating P38 MAPK to inhibit Nrf2/HO-1 signaling pathway [287]. β-elemene can be used as an inducer of iron apoptosis and the combination of cetuximab and β-elemene can be sensitive to KRAS mutant CRC cells by inducing ferroptosis and inhibiting EMT [48].

In addition, ferroptosis is thought to be involved in anti-tumor immune responses. For example, in melanoma models, CD8+ T cells modulate tumor ferroptosis during immunotherapy, which contributes to the anti-tumor efficacy of immunotherapy [288]. Dihydroartemisinin (DHA) is an active metabolite of artemisinin. Immune-stimulating nanodrugs (DHA and oxaliplatin) have been shown to induce ROS production in tumor cells, resulting in immunogenic programmed death and enhanced checkpoint blocking immunotherapy [289]. Recent literature suggests that DHA and pyropheophorbide-iron can produce and aggregate ROS to induce ferroptosis in an immunogenic manner, sensitizing non-immunogenic CRC, and can enhance the therapeutic effect of CRC anti-PD-L1 [290].

## 9. Tumor Microenvironment

In solid tumors, the TME consists of extracellular matrix, cancer-associated fibroblasts (CAFs), immune and inflammatory cells [291]. Components of the TME can act as therapeutic barriers for treating solid tumors [291]. The different cell types in TME are recruited to the tumor site in response to a series of therapeutic treatments that provide a sanctuary for tumor cells to escape drug-mediated cell death and develop resistance to tumor treatment drugs (Figure 3) [292].

### 9.1. Cancer-Associated Fibroblasts

CAF is a significant component of tumor stroma and expresses specific markers such as α smooth muscle actin (αSMA), fibroblast activation protein α (FAP) and fibroblast growth factor receptor (FGFR) [293]. In CRC, CAF-derived CCL2 upregulates FGFR4, which plays a role in tumor-matrix interactions and promotes EMT, which may constitute an essential factor in CRC resistance to 5-FU and oxaliplatin [294]. Transcriptional analysis in drug-resistant CRC cell lines showed that enhanced FGFR4 activity confers chemotherapeutic resistance, and FGFR4 silencing downregulation of anti-apoptotic proteins c-FLIP and Bcl-2 and decreasing STAT3 activity reverse the sensitivity of CRC to chemotherapy [50]. CAFs also mediate tumor resistance by secreting pro-inflammatory factors that alter tumor cell sensitivity and protect CRC cells from chemotherapeutic agents [295]. CAF can release TGF-β2 under hypoxia, synergize with HIF-1α, activate GLI2 expression through the non-canonical Hedgehog pathway and promote CRC resistance to 5-FU/oxaliplatin [296]. CAF-expressing Snail participates in CRC resistance to 5-FU/paclitaxel chemotherapy by secreting CCL1 to activate TGF-β and NF-κB signaling pathways, and inhibition of TGF-β and/or NF-κB signaling pathway can reverse CCL1-mediated multi-drug resistance [297]. Not only that, cytotoxicity itself activates CAF, which further promotes the self-renewal of CRC initiation cells, endowing CRC with chemotherapy resistance. This may be related to the secretion of interleukin-17A (IL-17A) induced by chemotherapy [298]. IL1β/TGFβ1 is a trigger for fibroblast recruitment and transformation into cancer-associated fibroblasts (CAF) in CRC and endows chemotherapy resistance [295]. However, the blocking of IL1β/TGFβ1 signaling did not increase sensitivity to chemotherapy, which may be a compensatory activation of the tak1-mediated non-canonical TGFβ pathway. Combined blocking of TGFβ -activated kinase 1 (TAK1) and TGFBR1 inhibits IL1β/TGFβ1-mediated fibroblast activation, reduces secretion of pro-inflammatory cytokines and makes tumor cells more sensitive to chemotherapy [295]. CAF-derived exosomes are also active agents of drug resistance. A series of studies have shown that CAF-derived exosomes, such as miR-92a-3p, H19 and CCAL, promote CRC EMT and lead to CRC metastasis and chemotherapy resistance by activating the Wnt/β-catenin pathway. Targeting these exosomes can effectively reduce autophagy activity and reverse CRC chemotherapy sensitivity [299,300,301].

During anti-EGFR therapy in CRC patients, CAF promotes tumor resistance to EGFR by secreting growth factors, such as FGF1, FGF2, HGF, TGF-β1 and TGF-β2 [302]. Conditional medium (CM) from CAF can confer anti-EGFR resistance in CRC stem-cell-like cells, which may be wild-type MET activated by CAF-derived HGF [303]. In addition, preclinical studies have demonstrated that FGFR4 overexpression secretes EGFR ligands, such as AREG, which activate EGFR and ERBB3, and FGFR4 overexpression reduces cetuximab-induced cytotoxicity. Compared with cetuximab, FGFR4 inhibitor (BLU9931) combined with cetuximab has a stronger anti-tumor effect [51].

Cell analysis of human and mouse CRC specimens showed that CAF is a major contributor to TGF-β production [39]. As previously mentioned, TGF-β activation in the TME can produce profound immunosuppressive activity against crucial cell types of innate and adaptive immunity. It has been demonstrated in a variety of tumor models, including CRC models, that activation of CAF may induce PD-1 and CTLA-4 expression on relevant immune cells, which leads to tumor insensitivity to immunotherapy or recurrence [39,304,305,306]. In addition, NADPH oxidase 4 (NOX4) is a common regulator of myofibroblast accumulation in many human cancers. NOX4 expression is closely related to CAF accumulation in myofibroblasts in patients with CRC, which may be attributed to the fact that NOX4 regulates the transdifferentiation from fibroblasts to myofibroblasts through the delayed period of intracellular ROS production, and the inhibition of NOX4 can effectively reduce the accumulation of CAF [53]. Preclinical studies have shown that MC38 models in CAF-rich mice have difficulty benefiting from tumor vaccines and anti-PD-1 therapy, possibly because CAFs exclude CD8+ T cells (not CD4+ T cells) from tumors by producing NOX4 T cells (or macrophages) to inhibit the response extensively, and pharmacological inhibition of NOX4 restores the immunotherapeutic response of CAF-rich tumors [306]. In addition, PD-L1 expression in CRC specimens is positively correlated with FGFR2 expression, which promotes PD-L1 expression in CRC by activating the JAK/STAT3 signaling pathway, which further promotes T cell apoptosis and leads to tumor immunosuppression. This phenomenon can be reversed by JAK/STAT3 signaling pathway inhibitors [233].

### 9.2. Myeloid-Derived Suppressor Cells

Myeloid-derived suppressor cells (MDSCs), a heterogeneous population of immature bone marrow cells, are one of the major immunosuppressive cell subpopulations in the tumor microenvironment and contribute to cancer immune avoidance by inhibiting the function of T and NK cells [307,308,309]. MDSCs are divided into three groups according to their histological characteristics: early or immature MDSC (E-MDSC/I-MDSC), mononuclear MDSCs (M-MDSCs) and polymorphonuclear/granulocyte MDSCs (PMN-MDSCs/G-MDSCs) [309,310]. The frequency of G-MDSCs was significantly higher than that of M-MDSCs in several CRC mouse models [308,311,312,313].

Immunosuppression is the main feature of MDSC in CRC TME. In the mouse CRC model, G-MDSCs expressed high arginase I (ARG1) in both peripheral blood and TME [313]; ARG1 depletes L-arginine, which maintains T cell proliferation and function, inhibits T cell proliferation and induces T cell apoptosis [314,315]. In addition, MDSC impairs Fc-receptor-mediated NK cell function by producing NO [316].

Cytokines play an important role in MDSC recruitment [317]. Preclinical studies have demonstrated that STAT1 recruits G-MDSC through IL-17 in colitis-associated colorectal cancer (CAC) models. Blockade of IL-17 reduced the recruitment of G-MDSC into intestinal specimen of STAT1 -/- mice and significantly decreased the expression of ARG1 and inducible nitric oxide synthase (iNOS) [317]. Other studies have reported that IL-8, GM-CSF, TNF-α, YAP1, CXCR2 and CCL2 promote G-MDSC recruitment in CRC and confer tumor immunotherapy resistance to CRC [308,311,318,319,320].

Inhibition of MDSC may be a promising way to reverse drug resistance to anti-PD-1. In a mouse colon-26 tumor model, inhibition of receptor tyrosine kinase KIT enhances the anti-tumor activity of immune checkpoint inhibitors (anti-CTLA-4 and anti-PD-1) by selectively reducing immunosuppressive M-MDSC clusters [54]. The humanized anti-Kit mAbs KTN0158 is currently being evaluated in patients with Kit-positive advanced solid tumors (NCT02642016). In addition, preclinical studies have demonstrated that TRAIL-R2 agonist antibodies can selectively target MDSC [321]. In a Phase I trial study, the efficacy of the novel agonist TRAIL-R antibody DS-8273A was evaluated, which caused a temporary reduction in the number of elevated MDSC in peripheral blood in most patients with advanced cancer, including CRC, to levels observed in healthy volunteers. In contrast, DS-8273A did not affect the number of neutrophil monocytes and other myeloid and lymphocyte populations (NCT02076451) [322].

### 9.3. Tumor-Associated Macrophages

Tumor-associated macrophages (TAMs) are one of the most abundant immune types in TME, influencing tumor progression by producing pro-inflammatory and anti-inflammatory cytokines [323]. In general, macrophages are divided into two subpopulations: classically activated, M1 macrophages and alternatively activated, M2 macrophages [323]. M1 macrophages (NOS2) are critical cellular components involved in inflammatory response and anti-tumor immunity. In contrast, M2 macrophages (CD163) play an anti-inflammatory and pro-tumor role, and TAMs often exhibit an “M2-like” phenotype [324]. Preclinical studies have shown that TAM (CD68) can be activated during 5-FU treatment and can give CRC cells against chemotherapy with 5-FU during 5-FU treatment, which may be caused by TAM secreting ornithine decarboxylase (ODC)-dependent decay activation JNK-caspase-3, which further leads to the CRC cells against chemotherapy with 5-FU [55]. Pharmacological or genetic blockade of ODC reverses TAM-induced chemotherapy resistance to 5-Fu in vitro and in vivo [55]. Data in the mouse MC38 tumor model suggest that TAM-derived milk-fat globule-epidermal growth factor-VIII (MFG-E8) mediates tumor chemotherapy resistance by activating STAT3 and Hedgehog signals in tumor stem cells [325]. In stage III CRC patients treated with 5-FU, L-OHP and leucovorin, high TAM (CD68) density is associated with treatment insensitivity. TAM-derived IL6 induces CRC chemotherapy resistance by activating the IL6R/STAT3/miR-204-5p axis, which can be reversed by miR-204-5P overexpression or IL6 blockers [326]. However, high densities of TAMs have been shown to significantly benefit stage III CRC patients treated with 5-FU [327]. It may be that 5-FU exposure is beneficial to the polarization of unpolarized macrophages towards M1, and that 5-FU and TAM (CD68) show a synergistic effect in increasing CRC cell death [327]. CD68 pan-macrophage marker does not allow subclassification of TAM, which may be responsible for the contradiction between prognosis and chemotherapy sensitivity. One instance of clinical data suggests that the M1/M2 ratio reflects the sensitivity of CRC patients to adjuvant therapy [328]. Therefore, the sensitivity of TAM to CRC chemotherapy depends on the M1/M2 ratio in TME.

TAM can mediate tumor immunosuppression by expressing immune checkpoints. In human and mouse CRC tumor models, high levels of PD-1 expression on TAM have low tumor phagocytosis, and treatment with checkpoint inhibitors targeting PD-1 or PD-L1 increases phagocytosis and reduces tumor growth [329]. Colony-stimulating factor 1 (CSF1) and its receptor CSF-1R play an essential role in the recruitment, differentiation and maintenance of immunosuppressive macrophages in tumors [330]. It has been shown in tumor-conditioned medium (TCM) differentiated macrophages that CSF-1 differentiated macrophages affect the frequency and the activity of T cells [56]. CSF-1R inhibition can specifically eliminate immunosuppressive M2-like macrophages and positively transfer the ratio of CD8 to CD4 to cytotoxic effector T cells [56]. An ongoing clinical study is evaluating the efficacy of anti-CSF1R (pexidartinib) in combination with anti-PD-L1 (durvalumab) in advanced/metastatic CRC or pancreatic cancer (NCT02777710). In addition, regorafenib also reduces TAMs by inhibiting the CSF-1R [331]. A combination of regorafenib and nivolumab showed activity in Japanese third-line or later pMMR mCRC with an ORR of 33.3% [332]. However, this strategy has not been proven effective in the North American population, with an ORR of 7.1% [333].

Therefore, TAM consumption or blocking cancer-induced M2-like macrophage programming has the potential to improve treatment efficiency and reduce anti-cancer drug resistance. Macrophages have been shown to be plastic [334,335,336]. Preclinical studies show that prostaglandin E2 (PGE2) receptor 4 (EP4), as the primary regulator of IMC, can mediate drug resistance by promoting the differentiation and proliferation of M2 TAM and MDSC [58]. TP-16 (EP4 antagonist) treatment polarized TAM from immunosuppressive M2 type to pro-inflammatory M1 type, which enhanced T-cell-mediated anti-tumor immunity and improved the efficacy of anti-PD-1 [58]. In addition, a series of studies have proved that *Ganoderma lucidum* spore polysaccharide (GLSP) [337], CMPB90-1 [338], natural M1 exosomes [339], conditional deletion of the microRNA (miRNA)-processing enzyme DICER [334] and TLR 7/8 agonist [57] can specifically eliminate M2-like phenotype or reset TAM from M2-like phenotype to M1-like phenotype to inhibit tumor development and reverse tumor treatment resistance.

### 9.4. Angiogenesis

Vascular endothelial growth factors (VEGFs) and their receptors (VEGFRs) are uniquely required to balance the formation of new blood vessels with the maintenance and remodeling of existing ones, during development and in adult tissues [340]. However, VEGF in the TME could drive immunosuppression by inducing vascular aberrations, inhibiting antigen-presenting and immune effector cells or enhancing the immunosuppressive activity of Tregs, MDSCs and TAMs [341]. In turn, immunosuppressive cells can drive angiogenesis, thereby creating a vicious cycle of suppressed anti-tumor immunity [341]. Bevacizumab is a humanized monoclonal antibody that binds to the VEGF-A isoform, thereby preventing the interaction of VEGF-A with VEGFR, thereby inhibiting the activation of the VEGF signaling pathway that promotes neovascularization [342]. Preclinical studies have demonstrated that VEGF blockade can reverse T cell failure and enhance the therapeutic effect of ICI [59,60]. However, in clinical studies, only patients with MSI-H CRC benefited from the combined blockade of VEGF and PD-1. For example, in a Phase Ib trial study evaluating the efficacy of a combination of atezolizumab and bevacizumab in MSI-H CRC patients, the disease control rate was 90% [84]. However, in another clinical study, the addition of atezolizumab did not improve survival compared with microsatellite-stable (MSS) CRC treated with bevacizumab and capecitabine [85]. Similar results were obtained in the NSABP C-08 study [343]. Since MSI-H tumors are hypermutated and highly immunogenic, they should evade the attack of the immune system in order to progress, and VEGF may be one of the critical molecules mediating immune evasion. Therefore, using VEGF inhibitors while inducing more immunogenicity of MSS tumors may be a promising approach. Phase 2 study results of CheckMate 9X8 show nivolumab plus standard of care (SOC, fluorouracil/leucovorin/oxaliplatin/bevacizumab) This shows higher response rates in first-line treatment of mCRC compared to SOC [344].

High expression of VEGF is also associated with drug resistance to EGFR therapy. Cetuximab can reduce VEGF expression, and high VEGF expression is associated with lower response rates and shortened PFS in mCRC patients treated with cetuximab [345]. Therefore, anti-VEGF mAbs may reverse EGFR resistance in mCRC patients. In a mouse CRC xenograft model, combined blocking of VEGF and EGFR significantly inhibited tumor growth and angiogenesis [61]. However, in another preclinical study, the combination of bevacizumab and cetuximab did not increase apoptotic tumor cell death compared to either drug alone [62]. In a clinical study, a combination of cetuximab and regorafenib was used in 17 mCRC patients who had previously received both anti-VEGF and anti-EGFR therapy. Eight mCRC patients showed clinical benefit, including partial response in one patient and stable disease in seven [86].

## 10. Conclusions

Although there is use of multiple agents in clinical practice, such as chemotherapy, targeted therapy and immunotherapy, the emergence of drug resistance is the most critical factor in the poor prognosis of CRC. Tumor cells can protect themselves from treatment effects by changing drug metabolism, drug transport, drug targets and cell death pathways. In addition, changes in these pathways can also mediate drug resistance by controlling several basic signaling pathways involved in drug resistance. Recently, more and more attention has been paid to the role of TME in drug resistance. We reviewed the data to show that changes in pathways from drug metabolism to eventual cell death play a key role in CRC cell drug resistance. Although many potential therapeutic targets for these pathways are described in our review, most of the data are based on limited preclinical data, and larger, multi-institutional studies including more preclinical and clinical analyses are needed to assess clinical significance of these potential therapeutic targets. In addition, many preclinical studies have shown that multi-drug combination therapy may be an effective way to overcome drug resistance, and further clinical studies are urgently needed to evaluate the effectiveness of combination therapy in improving drug resistance.

## Figures and Tables

**Figure 1 cancers-14-02928-f001:**
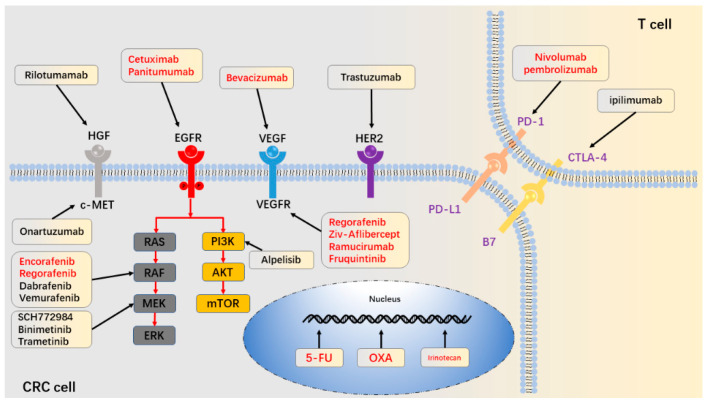
Currently FDA-approved (red) or promising (black) anti-CRC drugs. HGF: hepatocyte growth factor; c-MET: mesenchymal-epithelial transition factor; VEGF: vascular endothelial growth factor; EGFR: epidermal growth factor receptor; PI3K: phosphoinositide 3-kinase; AKT: protein kinase B, also known as PKB; mTOR: mammalian target of rapamycin; MEK: mitogen-activated protein kinase; ERK: extracellular signal regulated kinase; CTLA-4: cytotoxic T-lymphocyte-associated antigen 4; B7: B7 ligand; PD-1: programmed death-1; PD-L1: programmed death ligand 1; 5-FU: 5-fluorouracil; OXA: oxaliplatin.

**Figure 2 cancers-14-02928-f002:**
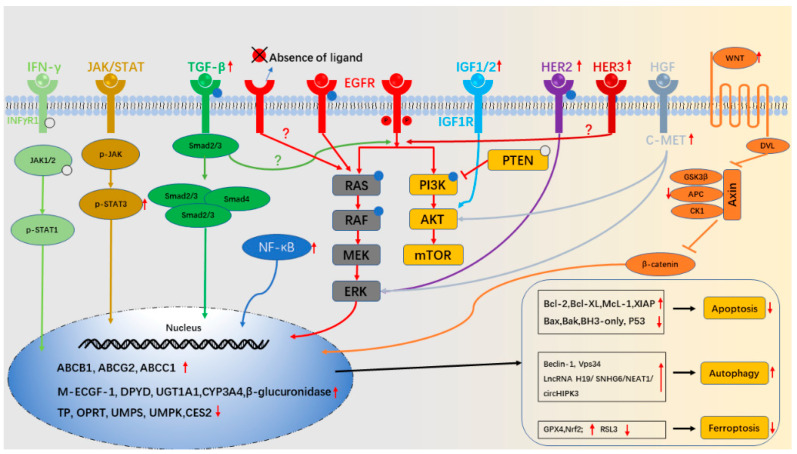
The changes of drug targets and signaling pathways lead to the development of CRC resistance. IFN-γ: interferon-γ; JAK/STAT: janus kinases/signal transducer and activator of transcription; PTEN: phosphatase and tensin homolog; APC: adenomatous polyposis coli; TP: thymine phosphorylase; OPRT: orotate phosphoribosyl-transferase; UMPS: uridine monophosphate synthetase; UMPK: UMP kinase; CES2: carboxylesterase 2; ABCB1: multi-drug resistance protein 1, MDR1, P-gp; ABCG2: breast cancer resistance protein, BCRP; ABCC1: multi-drug-resistance-associated protein 1, MRP1; UGT1A1: uridine diphosphate glucuronosyltransferase 1A1; CYP3A4: cytochrome P450-3A4.

**Figure 3 cancers-14-02928-f003:**
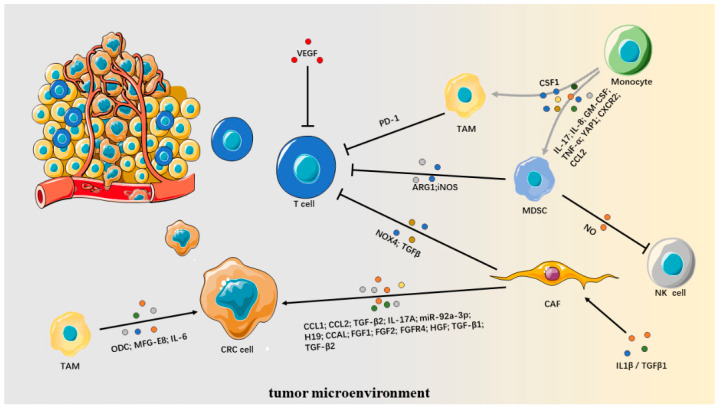
The changes of the TME lead to the development of CRC resistance. TAM: tumor-associated macrophages; MDSC: myeloid-derived suppressor cell; CAF: tumor-associated fibroblasts; ODC: ornithine decarboxylase; CSF1: colony-stimulating factor 1; NK cell: natural killer cell; TGF-β: transforming growth factor-β; IL-6: interleukin-6; FGF: fibroblast growth factor; ARG1: arginase I; iNOS: inducible nitric oxide synthase; NO: nitric oxide.

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
