# Peer review of "Drug Resistance in Colorectal Cancer: From Mechanism to Clinic"

_cancers, 2022, doi:10.3390/cancers14122928_

Round 1

Reviewer 1 Report

The paper aims a difficult and complex topic of an overview of the mechanisms involved in the resistance to the colon cancer treatment.

Due to the wide range of topics proposed, it is quite difficult to treat the subject only within a review. I think that for such a topic it is necessary to make a book. Due to the diversity of topics covered, some information may be omitted, especially recent studies.

On the other hand, the paper is well constructed, clear, with an explanation of the mechanisms and the presentation of studies on each mechanism. Even if some more recent studies have emerged, they need to be verified over time and argued for or against in other future studies.

I have an observation regarding the CheckMate 9X8 study (line 909) which failed to meet its primary end point of progression-free survival vs standard-of-care alone, however, it showed a higher response rate in first line treatment of metastatic colorectal cancer.

To increase the value of the material, where possible, I recommend updating the data instead of "future studies are needed." I would leave this expression only in the conclusions.

Reviewer 2 Report

Colorectal cancer is a very aggressive cancer, difficult to cure and diagnose, making it one of the leading causes of death. Besides the high incidence of colorectal cancer, most of the current therapies end up with development of resistance. In general, resistance mechanisms are nowadays a big challenge for cancer therapy and cure. Drug resistance occurs via different mechanisms, including changes in drug metabolism and transport, modification of the drug target, activation of bypass pathways to overcome drug treatment, or downregulation of cell death pathways.

The manuscript from Wang et al. is a comprehensive review of the different ways cells can escape drug treatments and become resistant. Drug resistance is a relevant topic, and this review puts together the molecular mechanisms of resistance and the current clinical data. The manuscript is well-structured and easy to read. For each specific pathway of drug resistance, the knowledge gap is clearly identified in each paragraph.

The figures are clear and easy to interpret. They distinctly reflect the information and the order of the text.

Many clinical and preclinical data are referenced to support the statements, and they are up to date. The tables can be a helpful tool to have an overview of the CRC drug resistance mechanisms and find the correct reference to either established clinical data or current clinical trials.

Overall, the manuscript would be helpful as an updated review of the drug resistance pathways specific for colorectal cancer. This paper would be of interest to clinicians but also for basic molecular biology research. It is suitable for publication after minor revision.

Major points:

·       Key references supporting broader statements are missing. These citations are usually missing from the first sentences of each paragraph, when concepts are introduced. It is important that the authors support the introductive concepts by citing the most relevant papers.

·       Moreover, the authors should make sure that the acronymous are correctly spelled before using them in the text.

·       The authors sometimes describe a pathway of drug resistance without mentioning what the drug does. It is important to describe first how the drug works and then move to the escape mechanism.

Minor comments:

·       Table 1, the column “species” should be changed with “model system”. Cells and PDX are not species.

·       “Abnormalities” should be replaced with a different word throughout the text because it is misleading.

·       Line 196. Replace library with pool.

·       Line 216. Replace coated with bound.

·       Line 646. Remove P value. This was never shown for other data, and it is not relevant.

·       Consider changing the title in “drug resistance in colorectal cancer: from mechanisms to the clinic”. The strength of this manuscript relies in the connection between the molecular pathways and the clinical data. It should be explicit in the title.
